# Limited changes in locomotor recovery and unaffected white matter sparing after spinal cord contusion at different times of day

**Lukasz P. Slomnicki**[1,2], **George Wei**[1,2,3,4], **Darlene A. Burke**[1,2], **Emily R. Hodges**[1,2], **Scott A. Myers**[1,2], **Christine D. Yarberry**[1,2], **Johnny R. Morehouse**[1,2], **Scott R. Whittemore**[1,2,3,5], **Sujata Saraswat Ohri**[1,2], **Michal Hetman**[1,2,3,5]*

1 Kentucky Spinal Cord Injury Research Center, University of Louisville School of Medicine, Louisville, Kentucky, United States of America, 2 Department of Neurological Surgery, University of Louisville School of Medicine, Louisville, Kentucky, United States of America, 3 Department of Pharmacology & Toxicology, University of Louisville School of Medicine, Louisville, Kentucky, United States of America, 4 University of Louisville School of Medicine, Louisville, Kentucky, United States of America, 5 Department of Anatomical Sciences & Neurobiology, University of Louisville School of Medicine, Louisville, Kentucky, United States of America

* michal.hetman@louisville.edu

**Data Availability Statement:** All relevant data are within the manuscript and its Supporting information files.

## Abstract

The circadian gene expression rhythmicity drives diurnal oscillations of physiological processes that may determine the injury response. While outcomes of various acute injuries are affected by the time of day at which the original insult occurred, such influences on recovery after spinal cord injury (SCI) are unknown. We report that mice receiving moderate, T9 contusive SCI at ZT0 (zeitgeber time 0, time of lights on) and ZT12 (time of lights off) showed similar hindlimb function recovery in the Basso mouse scale (BMS) over a 6 week post-injury period. In an independent study, no significant differences in BMS were observed after SCI at ZT18 vs. ZT6. However, the ladder walking test revealed modestly improved performance for ZT18 vs. ZT6 mice at week 6 after injury. Consistent with those minor effects on functional recovery, terminal histological analysis revealed no significant differences in white matter sparing at the injury epicenter. Likewise, blood-spinal cord barrier disruption and neuroinflammation appeared similar when analyzed at 1 week post injury at ZT6 or ZT18. Therefore, locomotor recovery after thoracic contusive SCI is not substantively modulated by the time of day at which the neurotrauma occurred.

## Introduction

Time of day affects the incidence of various acute pathologies including myocardial infarct (MI) and ischemic stroke [1]. Moreover, the severity of those injuries may also be influenced by the time of day at which they occur [2, 3]. Such effects stem from circadian rhythmicity of biological processes that determine the risk of a blood vessel occlusion and/or rupture and/or modify the tissue injury response [1]. Thus, in morning hours when the human active period begins, circadian maxima (acrophases) of blood pressure, sympathetic tone, and hemostasis may explain the higher occurrence and greater severity of MI and stroke seen at that time [1]. In addition, circadian modulation of metabolism, pro-inflammatory potential, immunity,

**Funding:** Kentucky Spinal Cord and Head Injury Research Trust (contract# 18-2, multi-PI award, communicating PI: MH, co-PI: SSO), NS108529 (multi-PI award, communicating PI: SRW, co-PI: MH), NS114404 (multi-PI award, communicating PI: MH, co-PIs: SRW and SSO), Norton Healthcare, and the Commonwealth of Kentucky Challenge for Excellence. The funders had no role in study design, data collection and analysis, decision to publish, or preparation of the manuscript. Salary support from the funders was as follows: Kentucky Spinal Cord and Head Injury Research Trust (LPS), NS108529 and NS114404 (LPS, SAM, CDY, JRM, DB, MH, SRW, SSO), Norton Healthcare (MH and SRW), the Commonwealth of Kentucky Challenge for Excellence (MH and SRW).

**Competing interests:** The authors have declared that no competing interests exist.

anti-oxidant defenses and blood-tissue barriers may directly affect the sensitivity to acute injuries [1, 4–6].

Circadian rhythmicity of biological processes is mediated by oscillations of gene expression produced by a set of conserved transcription factors (TFs) of the clock pathway including BMAL1, CLOCK and NPAS2 [7]. The clock pathway in hypothalamic suprachiasmatic nucleus (SCN) neurons synchronizes clock pathways in other cells of the body. The clock pathway TFs engage feedback loops that underlie oscillating expression of them and their regulators.

In rodents, the activity of the effector outputs of the clock pathway in most non-SCN tissues is low at ZT18-0 (late night/early morning) or high at ZT6-12 (afternoon/early evening), respectively [8–10]. Such oscillations coincide with differential responses to such challenges as MI, infection, endotoxic shock or autoantigen exposure while genetic disruption of clock signaling nullifies those time of day effects [11–15].

Time of day effects are also documented in several models of acute brain injury [16–21]. However, the maximum severity of brain damage peaked at distinctly different times depending on the model used [16–21]. Such variability suggests that unique, injury-specific pathogenic mechanisms may be differentially sensitive to circadian regulation. There are no reports of circadian effects on spinal cord injury (SCI).

BMAL1 is the principal non-redundant TF of the clock pathway output [7]. After moderate contusive SCI at the T9 level, $Bmal1^{-/-}$ mice showed enhanced locomotor recovery, increased white matter sparing as well as reduced inflammation and improved blood-spinal cord barrier function in the injury epicenter region [22]. Therefore, the pathogenesis of SCI may be regulated by circadian rhythms. The current work was initiated to test whether SCI outcomes differ if the injury occurs at different times of the day.

## Methods

### Animals

Six-week old C57Bl/6 wild-type female mice were obtained from the Jackson Laboratory (Bar Harbor, ME). Animals were maintained in a 12:12 light-dark cycle (6:00 light on, 18:00 light off) with food and water available *ad libitum* for 2 weeks. After five days of habituation to handling (performed in the same room where behavioral assessments were later performed) mice were randomly assigned to different experimental groups. All animal procedures were approved by the University of Louisville Institutional Animal Care and Use Committee and strictly adhered to NIH guidelines.

### Spinal cord injury

Avertin anesthesia, T9 spinal cord contusion (50 kdyn, IH impactor, Infinite Horizons, Lexington, KY) and post-surgery care were performed as previously described [22, 23] (see S1 Methods for detailed information including anesthesia and post-surgery analgesia). The surgeries were performed in two separate studies by the same team of investigators 12 h apart. In study 1, surgeries were at ZT0-1.5 (6:00–7:30, n = 11) or ZT12-13.5 (18:00–19:30, n = 11); in study 2, surgeries were at ZT5.5–6.5 (11:30–12:30, n = 14) and ZT17.5–18.5 (23:30–00:30, n = 14). Both groups were given identical post-surgical care and maintained under the same conditions for six weeks. Three mice were lost (1 in ZT0 euthanized after accidental rupture of the bladder during bladder expression, 2 in ZT12 were found dead at dpi 8 and dpi 10). See S1 Table for detailed contusion parameters (recorded force, displacement, velocity).

## Assessments of locomotor function

All behavioral assessments were performed at the same time (9:00–11:30) for both groups of mice by individuals without knowledge of group assignment. Hindlimb locomotor function was evaluated in an open field using the Basso Mouse Scale (BMS) by raters trained by Dr. Basso and colleagues at the Ohio State University [24]. Evaluations were performed weekly, first before the injury to determine baseline values, and then for six weeks starting at week 1 after SCI. The horizontal ladder test was performed as described previously using Columbus Instruments Sensor and RS-232 Mini Counter (Columbus Instruments; Columbus, OH, USA) with 2.5 mm rungs spaced 3.5 cm apart [25]. Briefly, each animal underwent five stepping trials per session and the total number of footfalls was quantified for the left and right limbs, respectively. A baseline session before SCI was followed by bi-weekly assessments starting at 2 weeks post-injury. To normalize the highly variable individual performance that was observed in this test as assessed by the number of errors at each post-injury testing time, the difference in the number of errors between the current and the previous testing session was also calculated for each animal. The positive or negative value of that parameter indicates worsened or improved ladder walking test performance as compared to the previous testing session, respectively. Gait was analyzed using the Treadscan Gait Analysis system (Cleversys, Reston, VA) after completion of all locomotor assessments at week 6 after SCI as described previously [26]. Briefly, mice were placed on a variable speed treadmill with a clear belt, and a high-speed digital video (ventral view) of the stepping animals was recorded. Each animal was placed on the treadmill, the speed of the belt increased until the animal could no longer maintain position, and then decreased slowly until stable walking was achieved. Ten+ consecutive step cycles were recorded at such individually optimized walking speed (if an animal was unable to execute 10 cycles, at least two 5+ step cycles were recorded). A minimum of 8 step cycles/animal were then analyzed using the Treadscan software.

*White matter sparing* was performed as described previously [23]. Briefly, after completion of behavioral analyses (day 42 post-injury), mice were deeply anaesthetized and perfused transcardially with ice cold phosphate buffered saline (PBS) followed by 4% paraformaldehyde (PFA) in PBS. Twenty μm serial transverse cryosections from a 4 mm spinal cord segment centered at the injury epicenter were stained for myelin with iron- eriochrome cyanine (EC). For each animal, the section with the least amount of myelinated white matter was identified as the injury epicenter. White matter sparing was defined as % relative white matter area (per total section area) at the epicenter as compared to the relative white matter area 2 mm rostral from the injury epicenter.

## Immunofluorescence staining for markers of blood-spinal cord barrier permeabilization

Some mice from study 2 (SCI at ZT6 and ZT18) were deeply anesthetized and transcardially perfused with PBS and then 4%PFA in PBS at 1 week post injury. Tissue processing, immunofluorescence staining, imaging and image analysis followed previously described methods [22]. Briefly, fibrin/fibrinogen, hemoglobin, CD36 and CD45 were detected using the following primary antibodies: anti-fibrinogen (α chain) (rabbit, 1:150, Bioss Antibodies, bs-7548R), anti-hemoglobin α (HBA1+2, rabbit, 1:150 LifeSpan BioSciences, LS-C409143), anti-CD36 (mouse, 1:100, BD Pharmingen, 552544), and anti-CD45 (rat, 1:150, Millipore, CBL1326). Every fifth longitudinal section from each cord (5x 20 μm sections/animal) was immunostained for the indicated markers and photographed with a 10× objective (Nikon Eclipse Ti epifluorescent microscope) and stitched using Nikon Elements software during acquisition. Elements software was used to threshold baseline brightness and contrast identically for each image for all

quantitative object and field measurements. The lesion site was defined as 1,500 μm region that was centered in the injury epicenter and included the injury penumbra. It corresponds to a heterodomain that is a pathology-affected region exhibiting extravascular deposition and disorganization of vascular laminin as well as disrupted GFAP immunoreactivity [27, 28]. The total area of the lesion site positive for each marker was quantified by digital image analysis using the basic densitometric thresholding features of Elements software, similar to methods previously reported [28]. Threshold values were obtained and set for each marker and held constant for each image quantified. The percentage of the lesion site area positive for each marker was quantified. For each animal, at least 3 sections spanning the injury epicenter were analyzed and marker signal area was averaged and normalized to WT control values.

### qRT-PCR analysis of circadian oscillations of gene expression

Naïve mice (coming from the same batch of animals as that used for SCI study 1 and undergoing same handling habituation) were deeply anesthetized and transcardially perfused with ice cold PBS at ZT1 or ZT12 to collect a 5 mm segment of the thoracic spinal cord and the liver. Total RNA extraction, synthesis of cDNA and SYBR Green-based qPCR analysis using the ΔΔCt quantification method and *Gapdh* as a normalizer followed previously described methodology [22]. See S2 Table for primer information. The results were compared to publicly available data on oscillations of the clock pathway mRNAs in various tissues of 7–8 week old C57Bl6 mice including the brain stem, the cerebellum, and the liver (http://circadb. hogeneschlab.org/mouse) [8].

### Statistical analyses

Repeated measures ANOVA (RM ANOVA) followed by Bonferroni *post hoc* t-test for multiple comparisons was used for analyzing BMS and horizontal ladder locomotor recovery data. Gene expression, white matter sparing, and immunofluorescence data were analyzed using the non-parametric Mann-Whitney U test.

## Results

In various tissues including intact male or female rat spinal cord as well as male C57Bl6 mouse brain stem, cerebellum or liver transcript levels for most clock pathway mRNAs oscillate with a maximum amplitude at or around ZT0 and ZT12 [8, 29] (http://circadb.hogeneschlab.org/mouse, S1 and S2 Figs). Such oscillations indicate activity of the clock pathway, as its core components are also clock pathway-regulated at the transcriptional level [7]. Therefore, levels of selected clock pathway transcripts were analyzed at ZT1 and ZT12 in the intact spinal cord and the liver of naïve mice taken from the same cohort that was used for the SCI study 1. At ZT12, *Bmal1* decreased by 42.5% in comparison to ZT1 (Fig 1A). Consistent with increasing BMAL1 TF activity during the rodent inactive period, expression of several BMAL1 target genes including *Nr1d1*, *Nr1d2*, *Cry1*, *Per1*, *Per2*, *Per3*, and *Dbp* was higher by 25–65% at ZT12. At that time, *Bmal1* showed a 98% decline in the liver with several of its target genes showing strong increases by 50–90% (Fig 1B). These data validate natural modulation of the clock pathway in the spinal cord and liver tissues of mice that were used for SCI experiments. The findings are consistent with greater circadian regulation of the transcriptome in the liver as compared to the CNS [8] and the reported maximum amplitude for many clock pathway genes at the start and the end of the mouse active period (S1 and S2 Figs).

To determine whether oscillations of the clock pathway activity at the time of injury correlate with a long-term locomotor recovery, moderate T9 SCI was first performed at ZT0 or ZT12 (study 1). Mean displacement was similar for both groups (567.3 ± 88.7 at ZT0 or

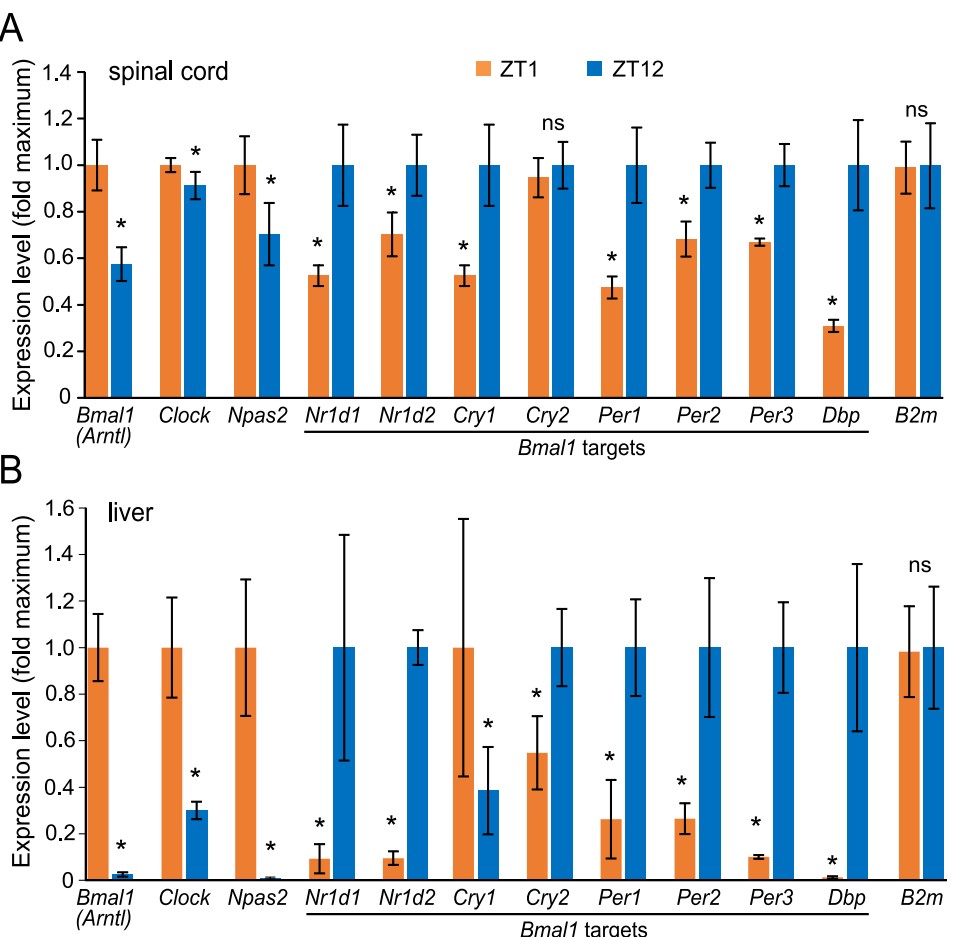

**Fig 1. Time of day effects on expression of clock pathway mRNAs in the intact mouse spinal cord.** Levels of mRNAs were determined at ZT1 and ZT12 by qPCR using total RNA from the lower thoracic segment of the spinal cord (A) and the liver (B). *Gapdh* was used as a normalizer for expression level determinations, *B2m* was also included as an additional normalizing transcript. Note that lower levels of *Bmal1* expression at ZT12 coincide with increased levels of several BMAL1 target genes suggesting increased activity of the clock pathway output. The observed differences in the spinal cord are consistent with reported maximal amplitudes of clock pathways mRNA in other non-SCN regions of mouse brain at the beginning and the end of the active period (S1 and S2 Figs). For each transcript, data represent average fold change of a time point with maximal expression ± SD; *, p<0.05, ns>0.05, U test; n = 3 mice/time point.

575.0 ± 83.1 μm at ZT12, p>0.05, *t*-test, S1 Table) suggesting no difference in severity of the primary injury. Similar recovery of hindlimb function was revealed with terminal BMS scores of 4.90 ± 0.61 or 4.83 ± 0.32 (Fig 2A). While mean number of errors in the ladder walking test was similar for both groups throughout the recovery, mean difference in error number between the current and the previous testing session suggested minor differences in performance trajectories (Fig 2B and 2B'). Thus, after initial improvement at week 4, ZT0 mice modestly declined at week 6 (error number difference -6.9±3.6 vs. 2.5±3.0, respectively; p<0.001, Bonferroni post-hoc t-test, Fig 2B'). In contrast, no significant changes were observed in the ZT12 group (error number difference -3.7±4.7 vs. -2.5±4.0, respectively; p>0.05, Bonferroni *post-hoc* t-test, Fig 2B').

In rodents with low thoracic level contusive SCI, white matter loss at the injury epicenter is the primary correlate of hindlimb functional deficits [24, 30]. Therefore, the observed limited

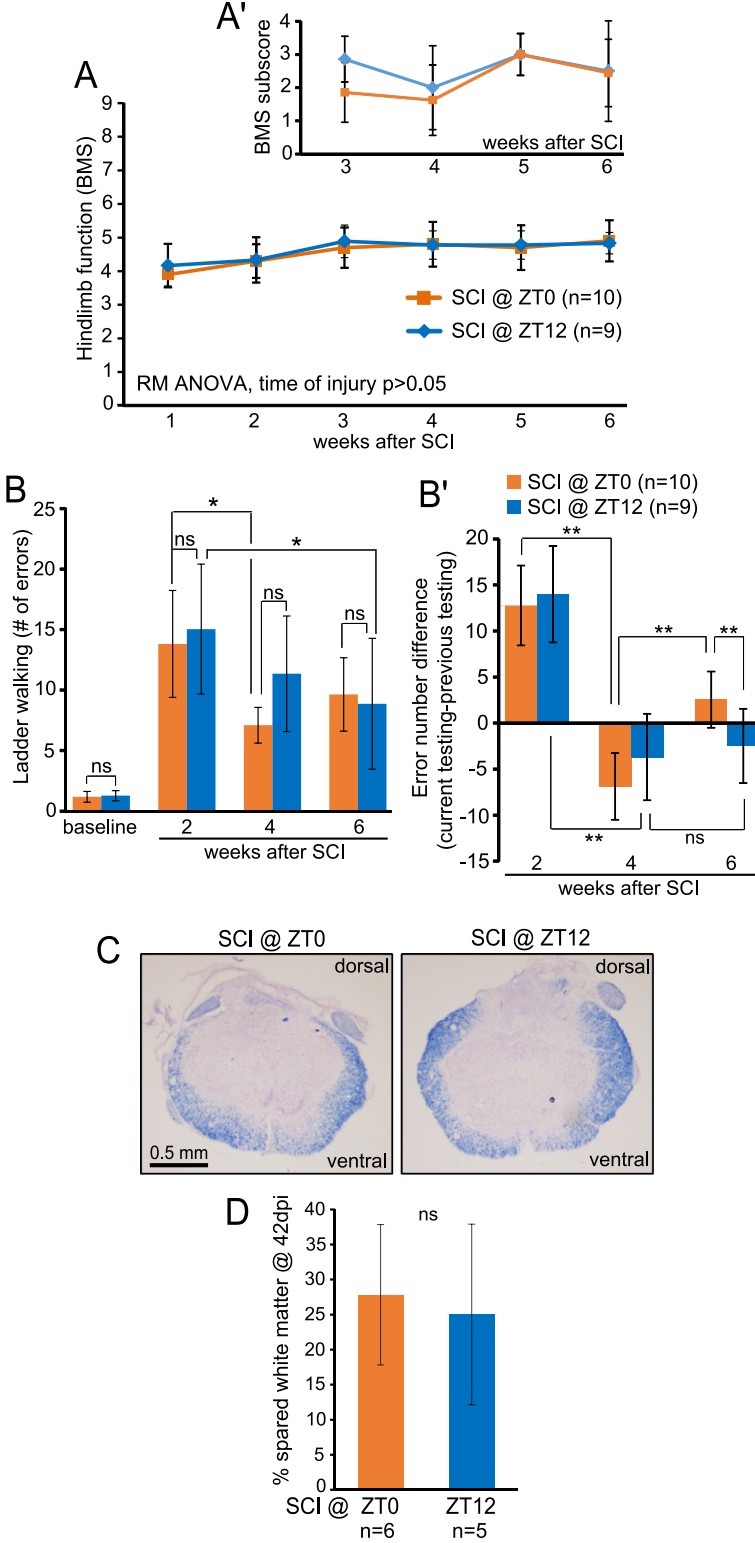

**Fig 2. Moderate effects on locomotor recovery and unaffected white matter sparing after SCI at ZT0 or ZT12. *A*, *B*,** Locomotor recovery after T9 50 kdyn IH contusion was quantified using the BMS (A and A') or the horizontal ladder test (mean number of errors is shown in B and mean difference in number of errors between the current and the previous testing session is shown in B'; for week 2 post-SCI the pre-injury baseline served as a normalization reference). The surgeries were done 12 h apart and all the assessments were done at the same time. For either

parameter, RM ANOVA showed significant effects of time after injury; time of injury only affected the ladder test performance but not the BMS (see S3 Table for more results). *C, D*, WMS was analyzed after completion of behavioral assessments 6 weeks after SCI. *C*, Representative images of EC myelin staining in spinal cord sections cut through the injury epicenter. *D*, Similar % spared white matter in the injury epicenter of ZT0 SCI and ZT12 SCI mice (ns, p>0.05, U test). All data represent means ± SD; in *B*, ns, *, **, *** represent p>0.05, <0.05, <0.01, <0.001, Bonferroni *post-hoc* t-test, respectively.

effects on functional recovery correlated well with the lack of significant differences in % spared white matter between the groups (Fig 2C and 2D).

Similar results were obtained in study 2 which compared effects of SCI at ZT6 and ZT18 (Fig 3). The only difference was a significantly higher number of ladder crossing errors in ZT6 vs. ZT18 mice at week 6 (18.68±4.71 vs. 12.52±3.07, respectively, p>0.01, Bonferroni *post-hoc* t-test, Fig 3B). Noteworthy, a minor yet significant difference at the baseline is unlikely to explain those week 6 effects as ZT18 mice were slightly worse performers before the injury (ZT6 vs. ZT18, 1.6±0.68 vs. 2.47±.57, respectively, p>0.01, Bonferroni *post-hoc* t-test Fig 3B). Analyzing individual error number difference from the previous session normalizes ladder test performance data against any potential baseline bias. Such analysis confirmed worsened performance of ZT6 mice at week 6 with no such worsening in ZT18 animals (Fig 3B'). The terminal kinematic analysis at week 6 revealed minor, yet significant, differences suggesting that ZT18 animals recovered more function as compared to ZT6 (S3 Fig). That improvement included slightly reduced compensatory usage of forelimbs and improved coordination of stepping without direct evidence for better hindlimb function (S3 Fig). Consistent with such a modest impact on functional recovery, similar % white matter was spared in ZT18 and ZT6 groups.

As it has been previously reported that in *Bmal1*$^{-/-}$ mice, acute/subacute (*i.e.*1-7 days post-injury) reductions in hemorrhage, blood spinal cord barrier (BSCB) disruption and neuroinflammation correlate well with enhanced recovery and terminal white matter sparing [22], study 2 also included several animals to evaluate those processes at 1 week post-injury. However, immunofluorescence analysis for markers of hemorrhage (hemoglobin), BSCB disruption (fibrinogen) or neuroinflammation (VCAM1, CD36, CD45) did not reveal any significant differences in their expression in the injury region of ZT6 vs. ZT18 mice (Fig 4). Therefore, even at the ZTs when the overall moderate effects on locomotor recovery appeared to be maximized, unaffected white matter sparing at week 6 correlated well with seemingly similar disruption of vascular integrity and the neuroinflammatory response 1 week after SCI.

## Discussion

Our results suggest that after contusive SCI, impairment of hindlimb function and its subsequent recovery are only modestly affected by the time of day at which the injury occurred. In addition, consistent with published clock pathway gene expression studies of rodent CNS tissue, we report relatively higher or lower activity of the clock pathway at the beginning and the end of the mouse active period, respectively [8, 9]. As reliable determination of the nadir and the zenith for a circadian-regulated mRNA requires probing at a minimum of 6 different time-points/day across at least two days, we do not have sufficient data to determine the spinal cord rhythm of clock pathway transcripts [31]. However, we can rely on published reports and/or publicly available data sets from other areas of the mouse CNS as well as other organs that, apart from SCN, show similar phase of oscillations for all major mediators of the clock pathway with a zenith of the clock pathway activity by the start of the active period and a nadir at its end (S1 and S2 Figs). In addition, our spinal cord or liver data show that clock pathway

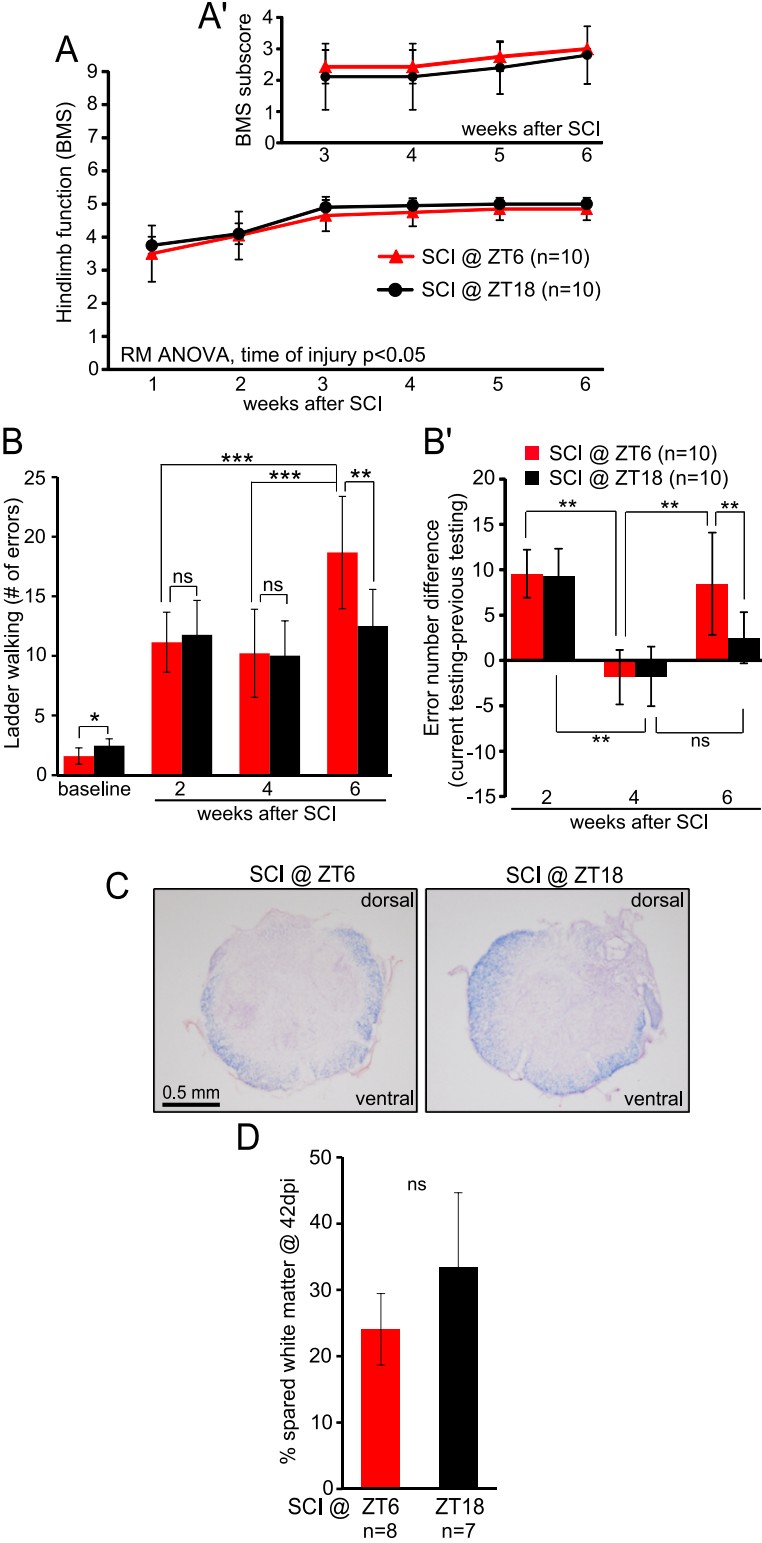

**Fig 3. Moderate effects on locomotor recovery and unaffected white matter sparing, after SCI at ZT6 or ZT18.** In study 2, SCI as well as functional and histological analyses were performed as described for study 1 (Fig 2). *A*, Despite the significant main effect in RMANOVA for time of injury, no significant group differences in BMS were observed with *post hoc* testing at any time point; BMS subscore (A') was also unaffected. *B*, horizontal ladder walking test revealed higher number of errors in ZT6 vs. ZT18 mice at week 6 which and a minor opposite difference at the

baseline; analyzing mean difference in number of errors between the current and the previous testing session as depicted in B' removed any potential baseline bias and still showed similar ZT6 group worsening at week 6 (B', see also S3 Table). *C*, Representative images of EC myelin staining in spinal cord sections cut through the injury epicenter. *D*, Similar % spared white matter in the injury epicenter of ZT6 SCI and ZT18 SCI mice (ns, p>0.05, *u*-test). All data are means ± SD; in *B*, ns, *, **, *** represent p>0.05, <0.05, <0.01, <0.001, Bonferroni *post hoc* t-test, respectively.

mRNA changes at ZT1 and ZT12 are of similar magnitude as maximal amplitudes reported in the mouse CNS or liver, respectively (Fig 1, S1 and S2 Figs). One notable exception is *Nr1d1/ Nr1d2* which peaks in the middle of the inactive period (S1 Fig). Therefore, our analysis likely underestimates its maximal circadian oscillations in the spinal cord and the liver. However, the presented findings confirm that in the cohorts of mice that were used for SCI studies, natural circadian oscillations of gene expression occurred in the spinal cord and the liver and that their phase was likely similar to that reported in rodents. Thus, circadian rhythmicity of gene expression and its effector biological processes could be considered as a potential variable that determines outcome of SCI.

While no significant changes in locomotor recovery were observed when using BMS to compare injuries at ZT0 vs ZT12 or ZT6 vs. ZT18, modest yet significant effects emerged in ladder walking test performance. At week 6, lower number of errors was observed in animals receiving SCI at ZT18 as compared to ZT6. Modest differences in recovery trajectories also emerged when comparing ZT0 vs. ZT12 mice. Overall, the relatively greater sensitivity of the ladder walking to detect time of day effects may be related to its dependence on additional components of locomotion such as hindlimb-forelimb co-ordination, sensory feedback, forelimb function, fine motor control of the digits and trunk stability [32]. Indeed, locomotion on a flat surface such as that assessed for hindlimbs in BMS is often unaffected when ladder test deficits are evident in various types of rodent CNS lesions [32]. Noteworthy, terminal kinematics analysis showed small, but significant, differences with lesser compensatory forelimb usage and improved coordination without improvement in direct measures of hindlimb function in ZT18 mice. Thus, more coordinated stepping and reduced need for compensatory activity of forelimbs could have contributed to the improved ladder test performance in that group. While mechanisms of those minor effects are unclear, they correspond well to unaffected terminal white matter sparing as that parameter most closely correlates with BMS [24, 30].

An intriguing possibility exists that the relatively better ladder test performance at week 6 in animals with night time vs. day time SCI may be due to reduced spasticity. The latter phenomenon, which is expected to have profound negative effects on ability to execute precise paw placements and grasp the ladder rungs, develops several weeks after rodent SCI reaching a plateau at around week 8 [33–35]. Of note, mice that were injured during the inactive period showed initial improvement in test performance from week 2 to week 4 that was followed by significant worsening at week 6 (Figs 2B and 3B). Such a time course would fit well with appearance of spasticity after rat or mouse SCI [33–35]. Interestingly, no significant week 6 worsening was observed in animals that were injured during the active period. Thus, different time of day at which injury occurs may affect development of spasticity during the chronic phase of recovery. Future studies would be needed to directly address such an interesting possibility.

While this report is the first to address the question of time of day effects on pathogenesis of SCI, others have documented existence of circadian modulation in other acute CNS injury models. Thus, in a middle cerebral artery occlusion (MCAO) model of rat stroke, acute infarct volume was three times larger with a stroke at ZT22 than at ZT10 [16]. However, relatively moderate time of day effects were reported in a murine MCAO model [21]. Global brain ischemia at ZT14 resulted in maximal hippocampal apoptosis in a rat, while the greatest

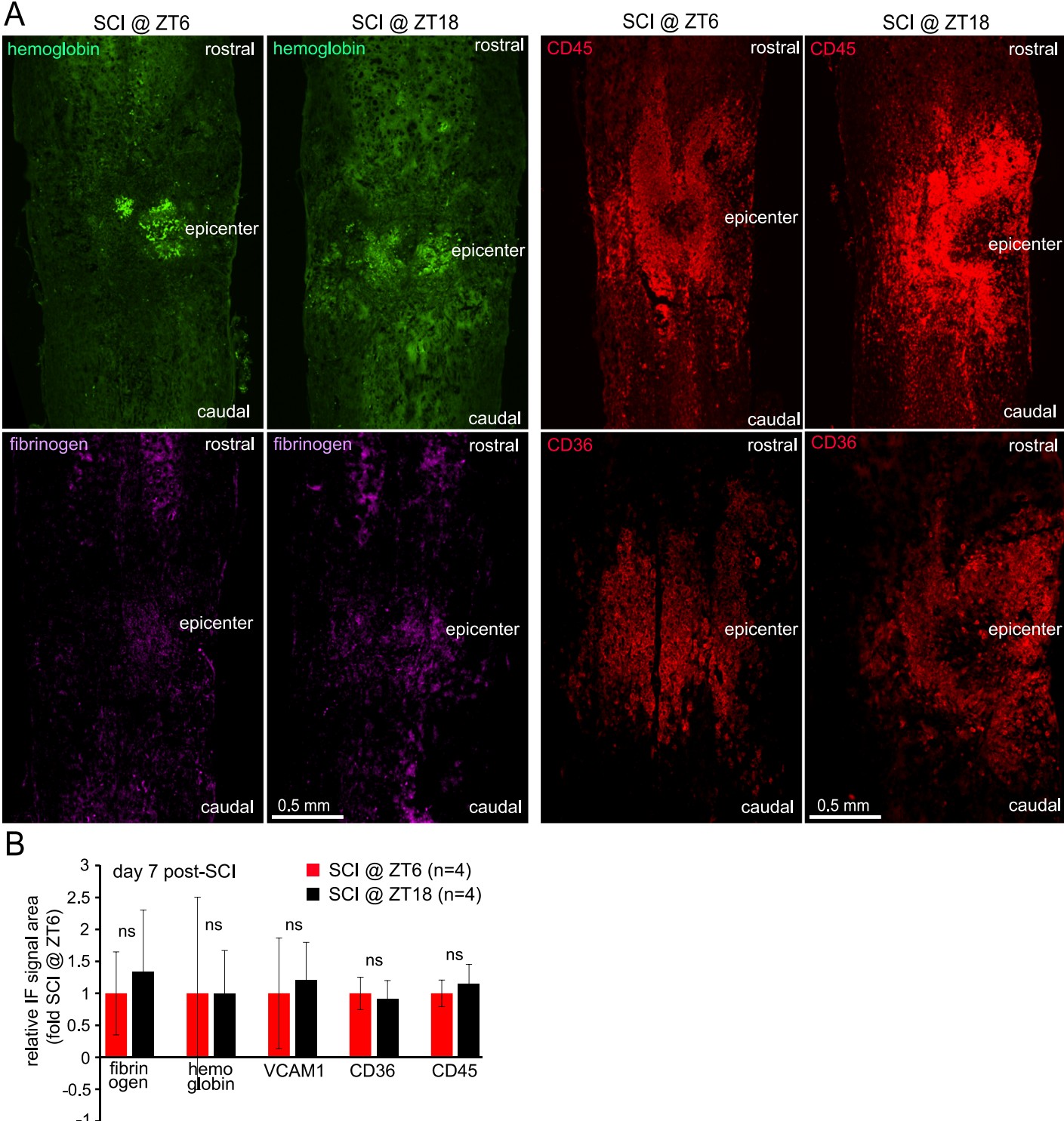

**Fig 4. Similar expression of markers of hemorrhage, BSCB disruption and neuroinflammation at 1 week after SCI at ZT6 and ZT18.** Some mice from study 2 were euthanized at 1 week post injury and markers of hemorrhage (hemoglobin), BSCB disruption (fibrinogen) and neuroinflammation (VCAM1, CD45 and CD36) were evaluated by immunofluorescence using longitudinal sections through the injury epicenter region. *A*, Representative images depicting marker stainings. *B*, Quantification of marker staining immunofluorescence (IF) signal area in a 1.5 mm lesion region spanning the injury epicenter (% total lesion area occupied by the positive IF signal). All data are means ± SD; ns, $p > 0.05$, U test).

hippocampal damage followed ZT6 ischemia in mice [17, 19]. In a mouse subarachnoid hemorrhage model, less apoptosis was observed in the hippocampus and the cortex when the insults occurred at ZT12 as compared to ZT2 [20]. Closed skull traumatic brain injury (TBI) in rats at ZT17 reduced brain damage area and acute mortality with a transient improvement in locomotor function as compared to ZT5 TBI [18]. These results indicate that dependent on the CNS injury model used, injuries that occurred in the second half of the active phase through the start of the inactive phase, when clock pathway activity reaches a nadir, produced maximal or minimal pathology at acute/subacute phases. While the currently presented SCI time of day studies did not reveal major differences in long term recovery or tissue sparing, the aforementioned reports of circadian effects focused on acute/subacute pathological changes with long term functional recovery or lesion healing either not examined or unaffected. Hence, in several types of acute CNS injury in rodents, including mouse SCI, the time of insult may have, at most, only transient effects on secondary damage with moderate long lasting functional impact.

One could argue that the severity of the SCI paradigm used for this work may have been too high to detect time of injury effects. This is unlikely as this moderate IH contusion has been used in several mouse studies in which locomotor recovery and terminal white matter sparing were increased or decreased by various genetic or pharmacological manipulations that targeted the secondary injury cascades [22, 23, 26, 28, 36, 37]. Hence, excessive primary damage is unlikely to explain our negative findings.

Why is post-SCI recovery only slightly affected by the time of injury despite our prior findings that $Bmal^{-/-}$ mice have improved functional outcome [22]? While there are many possible explanations to reconcile those observations, one should note that BMAL1 has been implicated in gene expression regulation beyond circadian rhythms [38, 39]. Loss of such a regulatory activity was proposed to contribute to a neurological phenotype of BMAL1-deficient mice and may have also played a role in SCI outcome [38]. Moreover, altered clock pathway/BMAL1 activity after, but not before, and/or at the time of SCI may be a critical contributor to secondary injury and long-term recovery after SCI. Interestingly, acute increases of at least some clock pathway components were observed in the injured mouse spinal cord tissue [22]. Those findings may indicate that after SCI, the clock pathway is reset to a new, post-injury time. Hence, modulation of the secondary damage by the injury-regulated activity of the clock pathway may override any earlier influences from natural circadian oscillations. Lastly, a confounding factor that may alter the significance of natural clock pathway oscillations in experimental SCI may be pre- and post-surgery care including such potential clock-resetting stimuli as anesthesia and analgesia [40, 41].

Noteworthy, a recent study using various rodent brain ischemia models showed that an acute CNS insult may engage different pathogenic mechanisms dependent on the time of day [42]. Specifically, active but not inactive period ischemia was shown to be associated with reduced penumbra and resistance to several established neuroprotective interventions suggesting differential diurnal engagement of potential therapeutic targets including reactive oxygen species or the NMDA receptor. Thus, while the final outcome of an acute CNS injury may be only moderately affected by the time of injury, the latter parameter may be an important determinant of responsiveness to neuroprotective interventions. Future studies are needed to examine such an interesting concept in the context of SCI.

## Conclusions

Current work using moderate contusive thoracic SCI indicates no major injury time of day effects on long term locomotor recovery or white matter integrity in mice. However, slightly

improved terminal performance in ladder walking after active period SCI suggests subtle, but significant, effects that may involve structural and/or functional plasticity of spinal cord circuitries. Obtaining clinical data to test if time of injury affects SCI outcome in humans will be extremely difficult as confounding injury co-morbidities and variable extents of spontaneous recovery will restrict necessary samples sizes [43, 44]. Future preclinical analyses to address potential effects on additional outcomes such as immune system dysregulation [45] or differential involvement of distinct pathogenic mechanisms of secondary injury [42] may provide insight into potential novel therapeutic avenues that could be acutely initiated depending on time of injury.

## Supporting information

**S1 Methods. Spinal cord injury and post-surgery animal care.**
(DOCX)

**S1 File. ARRIVE compliance questionnaire.**
(PDF)

**S1 Table. Contusion parameters for each individual animal.**
(DOCX)

**S2 Table. qPCR primers.**
(DOCX)

**S3 Table. Results of RM ANOVA for BMS and ladder test.**
(DOCX)

**S1 Fig. Circadian oscillations of selected clock pathway mRNAs in the brain stem, the cerebellum and the liver of C57Bl6 mice.** The data are from the publicly available circadian transcriptome database (http://circadb.hogeneschlab.org/mouse). All presented mRNAs show significant circadian oscillations in all three tissues (JTK $p<0.05$) except *Cry1* (non-significant cycling in the brain stem /JTK $p = 0.058$/) and *B2m* (no cycling in any tissue /JTK $p = 1$/). Note that the presented default output graphs from the circadb database show one full 24 h period from ZT24 though ZT48. Therefore, ZT1 or ZT12 in Fig 1 corresponds to ZT25 or 36 in S1 and S2 Figs, respectively.
(PDF)

**S2 Fig. Circadian oscillations of selected clock pathway mRNAs in the brain stem, the cerebellum and the liver of C57Bl6 mice.** The data are from the publicly available circadian transcriptome database (http://circadb.hogeneschlab.org/mouse). All presented mRNAs show significant circadian oscillations in all three tissues (JTK $p<0.05$) except *Cry1* (non-significant cycling in the brain stem /JTK $p = 0.058$/) and *B2m* (no cycling in any tissue /JTK $p = 1$/). Note that the presented default output graphs from the circadb database show one full 24 h period from ZT24 though ZT48. Therefore, ZT1 or ZT12 in Fig 1 corresponds to ZT25 or 36 in S1 and S2 Figs, respectively.
(PDF)

**S3 Fig. Minor differences in gait between mice injured at ZT6 vs. ZT18.** After completion of BMS and ladder walking at week 6 after SCI, gait analysis was performed using the Treadscan system. *A-B*, Although the average coordinated plantar index (CPI, a ratio between numbers of plantar step cycles with correct sequence of limb placement to all step cycles) was not different between the groups (A), significantly more ZT18 animals achieved high level of coordination (B, CPI>0.6). C-F, Stride analysis revealed no significant differences in

hindlimb function with consistent effects on forelimbs suggesting reduced compensatory usage in the ZT18 group. Those include longer and less frequent strides (C), as well as longer swings (D) and shorter stance (E). While the BMS-correlated rear track width was unaffected (Beare et al. 2009, PMID: 19886808), forelimb-hindlimb foot base of support was shorter in the ZT18 group (F). The latter parameter was the only direct indication of potentially improved hindlimb function in ZT18 mice (less hindlimb dragging); additional indirect support for moderate improvement in hindlimb function is provided by the aforementioned lower compensatory usage of forelimbs in that group. Data represent means ± SD; Binominal Proportion test, CPI distribution ($z = 2.0$, $p<0.05$, $*$); RM ANOVA with left vs. right side and time-of-injury used as two factors, effect of time-of-injury: swing ($F_{1,36} = 8.9$, $p<0.01^{**}$), % of swing ($F_{1,36} = 6.3$, $p<0.05^*$), % of stance ($F_{1,36} = 6.3$, $p<0.05^*$), stride ($F_{1,35} = 6.5$, $p<0.05^*$), stride frequency ($F_{1,36} = 6.3$, $p<0.05^*$), foot base of support ($F_{1,36} = 14$, $p<0.01^{**}$); ns, $p>0.05$. Overall, gait parameters suggest only minor effects of time-of-injury on locomotor recovery as compared at ZT6 vs. ZT18.
(PDF)

## Acknowledgments

We thank Jason Beare for assistance with behavioral analyses.

## Author Contributions

**Conceptualization:** Lukasz P. Slomnicki, Scott R. Whittemore, Sujata Saraswat Ohri, Michal Hetman.

**Formal analysis:** Darlene A. Burke, Emily R. Hodges, Scott A. Myers, Johnny R. Morehouse.

**Funding acquisition:** Scott R. Whittemore, Sujata Saraswat Ohri, Michal Hetman.

**Investigation:** Lukasz P. Slomnicki, George Wei, Darlene A. Burke, Emily R. Hodges, Scott A. Myers, Christine D. Yarberry, Johnny R. Morehouse.

**Supervision:** Michal Hetman.

**Writing – original draft:** Lukasz P. Slomnicki, Michal Hetman.

**Writing – review & editing:** Darlene A. Burke, Scott A. Myers, Johnny R. Morehouse, Scott R. Whittemore, Sujata Saraswat Ohri, Michal Hetman.

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
