## [Decision Letter · Decision Letter 0]

13 Apr 2021

PONE-D-21-09445

Unaffected functional recovery after spinal cord contusions at different circadian times

PLOS ONE

Dear Dr. Hetman,

Thank you for submitting your manuscript to PLOS ONE. After careful consideration, we feel that it has merit but does not fully meet PLOS ONE’s publication criteria as it currently stands. Therefore, we invite you to submit a revised version of the manuscript that addresses the points raised during the review process.

We look forward to receiving your revised manuscript.

Kind regards,

Simone Di Giovanni

Academic Editor

PLOS ONE

Journal Requirements:

Reviewers' comments:

Reviewer's Responses to Questions

**Comments to the Author**

1. Is the manuscript technically sound, and do the data support the conclusions?

Reviewer #1: Yes

Reviewer #2: Yes

2. Has the statistical analysis been performed appropriately and rigorously? 

Reviewer #1: Yes

Reviewer #2: Yes

3. Have the authors made all data underlying the findings in their manuscript fully available?

Reviewer #1: Yes

Reviewer #2: Yes

4. Is the manuscript presented in an intelligible fashion and written in standard English?

Reviewer #1: Yes

Reviewer #2: Yes

5. Review Comments to the Author

Reviewer #1: In this manuscript Slomnicki et al investigate whether a moderate T9 contusion SCI performed in mice at Zeitgeber time 1 or 12, when lights are turned on or off, effects the functional recovery and histological outcome. They confirm that several key clock regulators are differentially expressed at these times in uninjured spinal tissue. They then go on to perform a contusion injury at these two time points and demonstrate no difference in hindlimb recovery based on the BMS and ladder test as well as no difference in white matter sparing at the lesion epicentre.

The manuscript is well written and of interest to the field. The experiments are conducted to a high quality with clear aims and conclusions. I have only minor comments that need to be addressed before publication.

Minor concerns:

1. In the title the authors should to be careful when stating circadian time; this manuscript only investigates the effects of Zeitgeber time at two timepoints based around the light cycle.

2. There is often a lag before the circadian rhythm responds to changes in the expression of clock regulatory genes, it would be of interest to perform a SCI at other times points. However the reviewer appreciates that this may be out of the scope of the current manuscript.

3. In Supp. 1 and 2 the Zeitgeber time starts at 18, it would be more relevant to see the differences at 0 to 12 as these are the times used in the study.

4. It would be of interest to determine whether the injury resets the clock and induces changes in the expression of clock regulatory genes at the chronic time point.

5. In the authors previous publication using Bmal1-/- mice they demonstrate an effect on the integrity of the BSCB and inflammatory response acutely after SCI. It may be interesting to also examine some of these histological markers in this chronic SCI tissue.

6. In the methods please state at what time of day behavioural assessments were performed.

7. Supp.2 the Dbp plot needs to be re-aligned.

Reviewer #2: In the manuscript Slomnicki et al studied how the time of day affects functional recovery and anatomical adaptation after moderate contusive SCI at T9 level in mice, performing injuries at Zeitgeber time (ZT)1 and ZT12, i.e. when lights are on and off, respectively.

The authors confirmed differential expression of clock genes in uninjured spinal cord between ZT1 and ZT12. Then they performed SCI at these 2 time points and found no difference in hindlimb recovery as measured by BMS and ladder tests. Furthermore, they found no difference in sparing at the injury site in the white matter.

The manuscript is well written and the research question is of interest for the field. While the manuscript is experimentally of very good quality, the reviewer has a number of concerns/comments that need to be addressed before publication:

Major concerns:

1. Clock genes and their downstream pathways usually present a lag in the response to environmental stimuli (Zeitgebers such as light) as also shown by the clock gene expression in Suppl Fig 1 and 2. Indeed, while no difference was found between ZT0 and ZT12, increasing the number of injury time points, i.e. increased temporal resolution, is required to claim that time of day does not play a role, as the authors themselves point out in the discussion.

2. Since in their previous publication the authors found an effect on the integrity of the BBB and acute inflammatory response in mice with Bmal1 deletion, it would be relevant to examine these aspects in the present study. In fact, here there is no mention of the different cell types present in the intact and injured spinal tissue, what is the clock in the different cell types? And importantly, what happens to them before and after injuries performed at different ZTs?

3. Although no difference in functional recovery was found, circadian changes may be subtle and the analysis performed by authors not powerful enough to fully appreciate them, it may be relevant to analyse anatomical changes such as plasticity, regeneration and sprouting of different tracts/neurons affected by injury.

Minor comments

1. The title states “circadian times”, however the injuries are performed on mice kept in a normal LD cycle which makes it impossible to determine circadian time but only zeitgeber time or time of the day

2. In Suppl Fig 1 and 2 the graphs show a time starting at ZT18, it would be ideal to have them all starting at ZT0, especially considering that the injuries have been performed at ZT0 and ZT12

3. The time at which the lights are on is usually termed ZT0 and not ZT1. It is not clear why the author performed injury at ZT0 (Fig 2) but show qPCR analysis at ZT1 (Fig 1)

6. PLOS authors have the option to publish the peer review history of their article (what does this mean?). If published, this will include your full peer review and any attached files.

Reviewer #1: No

Reviewer #2: No

---

## [Author Response · Author response to Decision Letter 0]

20 Sep 2021

REVIEW RESPONSE: 

Reviewer #1, Point # 1: In the title the authors should to be careful when stating circadian time; this manuscript only investigates the effects of Zeitgeber time at two timepoints based around the light cycle.

Response: We replaced the term <circadian time> with either <zeitgeber time> or <time of day>. Also, additional injury timepoints were added for the revision (study 2 examines ZT6 vs. ZT18). 

Reviewer #1, Point # 2: There is often a lag before the circadian rhythm responds to changes in the expression of clock regulatory genes, it would be of interest to perform a SCI at other times points. However, the reviewer appreciates that this may be out of the scope of the current manuscript. 

Response: We fully agree that comprehensive analysis of time of injury effects should include more timepoints. Therefore, additional injury timepoints were added for the revision (the newly added study 2 examines ZT6 vs. ZT18, Revised Figs. 3 and 4). 

Reviewer #1, Point # 3: In Supp. 1 and 2 the Zeitgeber time starts at 18, it would be more relevant to see the differences at 0 to 12 as these are the times useTd in the study.

Response: the graphs in Figs. S1 and S2 were generated as a default output of the circadb database. Each graph covers one full period (ZT24-ZT48). Hence, ZT0 or ZT12 in our data (Fig. 1) corresponds to ZT24 or ZT36, respectively. To avoid confusion, we modified the legend for Fig S1 and S2 to explain that shift. 

Reviewer #1, Point # 4: It would be of interest to determine whether the injury resets the clock and induces changes in the expression of clock regulatory genes at the chronic time point.

Response: We agree that this would be an interesting question to answer. However, we feel that this resource-intensive experiment is out of scope of our current paper that is addressing the effects of time of injury on post-injury recovery rather than consequences of the injury for the rhythmicity (to do this correctly with injured animals one would have to study at least 2 periods with 4 probing times/period- 2x4= 8 timepoints per arm). In the context of our data, the most interesting time window for such an experiment would be during acute/subacute phases of recovery when white matter is still being lost and a clock reset could nullify any potential effects of time of injury on lesion size. At least with thoracic level injuries in humans or rats, rhythmicity of various physiological rhythms appears normal at chronic timepoints (PMIDs: 22474242, 21231876, 30627655). Thus, limited chronic outcome could be expected with T9 contusion in mice. 

Reviewer #1, Point # 5: In the authors previous publication using Bmal1-/- mice they demonstrate an effect on the integrity of the BSCB and inflammatory response acutely after SCI. It may be interesting to also examine some of these histological markers in this chronic SCI tissue.

Response: We agree that this would be an interesting question to answer. Unfortunately, we did not have any tissue left from our original study (study 1 with SCI at ZT0 and ZT12). However, we generated extra tissue in the new study that probed effects of SCI at ZT6 and ZT18. As in study 2, effects of injury time, while still limited, were greater than those in study 1 (ZT18 group showed small but significant improvement in BMS and better ladder test performance as compared to ZT6), we focused our BSCB/neuroinflammation analyses on that experiment. We have chosen day 7 post injury because at that timepoint BSCB leak is still present, neuroinflammation is well developed, and, Bmal1 deficiency strongly reduced both those processes (Slomnicki et al. Scientific Reports 2020). However, we did not observe any differences between SCI at ZT6 vs. ZT18. Those new data are shown in the revised Fig. 4. 

Reviewer #1, Point # 6: In the methods please state at what time of day behavioural assessments were performed.

Response: Modified as requested (all testing was done between 9:00-11:30).

Reviewer #1, Point # 7: Supp.2 the Dbp plot needs to be re-aligned.

Response: Modified as requested.

Reviewer #2, Major concern #1: Clock genes and their downstream pathways usually present a lag in the response to environmental stimuli (Zeitgebers such as light) as also shown by the clock gene expression in Suppl Fig 1 and 2. Indeed, while no difference was found between ZT0 and ZT12, increasing the number of injury time points, i.e. increased temporal resolution, is required to claim that time of day does not play a role, as the authors themselves point out in the discussion.

Response: We conducted another study that compared effects of SCI at ZT6 vs. ZT18. That experiment resulted in similar conclusions as the original experiment with SCI at ZT0 and ZT12. The new data are presented in the revised manuscript as Fig. 3 and Fig. 4. 

Reviewer #2, Major concern #2: Since in their previous publication the authors found an effect on the integrity of the BBB and acute inflammatory response in mice with Bmal1 deletion, it would be relevant to examine these aspects in the present study. In fact, here there is no mention of the different cell types present in the intact and injured spinal tissue, what is the clock in the different cell types? And importantly, what happens to them before and after injuries performed at different ZTs?

Response: We agree that defining injury time effects on acute/subacute dysfunction of BBB/BSCB and the neuroinflammatory response would be an interesting question to answer. Therefore, we generated extra tissue in the new study that probed effects of SCI at ZT6 and ZT18. As in that study (study #2), effects of injury time of day on recovery, while still limited, were greater than those in study #1 (ZT18 group showed small but significant improvement in BMS and terminal better ladder test performance as compared to ZT6), we focused our BSCB/neuroinflammation marker analyses on that experiment. We have chosen day 7 post injury 7 because at that timepoint BSCB leak is still present, neuroinflammation is well developed, and, Bmal1 deficiency strongly attenuates those processes (Slomnicki et al. Scientific Reports 2020). However, we did not observe any differences between ZT6 vs. ZT18. Those new data are shown in the revised Fig. 4.

While we agree that defining cell type specific regulation of the clock pathway in the intact and injured spinal cord would be of great interest, performing such a resource and time intensive analysis is beyond the scope of the current manuscript. 

Reviewer #2, Major concern #3. Although no difference in functional recovery was found, circadian changes may be subtle and the analysis performed by authors not powerful enough to fully appreciate them, it may be relevant to analyse anatomical changes such as plasticity, regeneration and sprouting of different tracts/neurons affected by injury.

Response: We agree that those additional analyses could provide more insight into subtle time of injury effects on the pathogenesis of SCI. However, such additional experiments would require significant resources and relatively long time to complete. Therefore, their addition would greatly delay timely publication of the current data. To acknowledge a possibility that such additional effects may exist, we added the following statement to the final conclusions paragraph of the discussion: 

<However, slightly improved terminal performance in ladder walking after active period SCI suggests subtle, but significant, effects that may involve structural and/or functional post-injury plasticity of spinal cord circuitries.>

Reviewer #2, minor concern #1: The title states “circadian times”, however the injuries are performed on mice kept in a normal LD cycle which makes it impossible to determine circadian time but only zeitgeber time or time of the day

Response: We replaced the term <circadian time> with <zeitgeber time> or <time of day>.

Reviewer #2, minor concern #2: In Suppl Fig 1 and 2 the graphs show a time starting at ZT18, it would be ideal to have them all starting at ZT0, especially considering that the injuries have been performed at ZT0 and ZT12

Response: The graphs in Figs. S1 and S2 were generated as a default output of the circadb database. Each graph covers one full period (ZT24-ZT48). Hence, ZT0 or ZT12 in our data (Fig. 1) corresponds to ZT24 or ZT36, respectively. To avoid confusion, we modified the legend for Fig S1 and S2 to explain that shift. 

Reviewer #2, minor concern #3: The time at which the lights are on is usually termed ZT0 and not ZT1. It is not clear why the author performed injury at ZT0 (Fig 2) but show qPCR analysis at ZT1 (Fig 1).

Response: We define ZT0 as the time of lights on (see the Abstract). The difference between the RNA analysis and the SCI study 1 was due to adjustments for staff availability. The rationale of the RNA analysis was to confirm oscillating activity of the clock pathway. That goal was met with the executed experimental design. Also, densely probed mRNA data from the circadb database (Figs. S1, S2), suggest that one hour difference between the experiments would unlikely change the overall conclusions as to the likely status of the clock pathway activity in the ZT0 SCI group.

---

## [Decision Letter · Decision Letter 1]

5 Oct 2021

PONE-D-21-09445R1Limited changes in locomotor recovery and unaffected white matter sparing after spinal cord contusion at different times of dayPLOS ONE

Dear Dr. Hetman,

Thank you for submitting your manuscript to PLOS ONE. After careful consideration, we feel that it has merit but does not fully meet PLOS ONE’s publication criteria as it currently stands. Therefore, we invite you to submit a revised version of the manuscript that addresses the points raised during the review process.

We look forward to receiving your revised manuscript.

Kind regards,

Simone Di Giovanni

Academic Editor

PLOS ONE

Journal Requirements:

Reviewers' comments:

Reviewer's Responses to Questions

**Comments to the Author**

1. If the authors have adequately addressed your comments raised in a previous round of review and you feel that this manuscript is now acceptable for publication, you may indicate that here to bypass the “Comments to the Author” section, enter your conflict of interest statement in the “Confidential to Editor” section, and submit your "Accept" recommendation.

Reviewer #1: (No Response)

Reviewer #2: All comments have been addressed

2. Is the manuscript technically sound, and do the data support the conclusions?

Reviewer #1: Partly

Reviewer #2: Yes

3. Has the statistical analysis been performed appropriately and rigorously? 

Reviewer #1: Yes

Reviewer #2: Yes

4. Have the authors made all data underlying the findings in their manuscript fully available?

Reviewer #1: Yes

Reviewer #2: Yes

5. Is the manuscript presented in an intelligible fashion and written in standard English?

Reviewer #1: Yes

Reviewer #2: Yes

6. Review Comments to the Author

Reviewer #1: Point 1: The new results for the normalised horizontal ladder in both study 1 and 2 appear confusing. The mice injured at ZT12 or ZT18 do not recover significantly more but rather that the mice injured at ZT0 or ZT6 get significantly worse at 6 weeks, do the authors have any suggestions in why these groups would deteriorate at week 6?

I can appreciate why the authors tried to normalized the horizontal ladder data due to high variability. However, this is an unusual and confusing way to analyse this data and I believe sufficient number of animals were used to observe real behavioural differences between the groups. I would recommend to show the unnormalized analysis using number of errors, as was provided in the first submission.

Point 2: On Line 264 the authors state “Both BMS (A) and normalized error score in the horizontal ladder walking test (B) revealed minor yet significant improvement in locomotor recovery between ZT18 and ZT6.” But the authors go on to say that “For BMS, no significant group differences were observed with post hoc testing at any time point” Please clarify and correct the statement and results of the BMS. The data appears to show an improvement with time, due to spontaneous recovery but no differences between the groups.

Point 3: Investigating the BSCB and neuroinflammation markers is a welcomed addition to the paper. It may also be interesting to assess fluorescence intensity of these markers at the lesion site. For instance, in the image in Fig.4 CD45 staining appears to be more intense at ZT18.

Point 4: The figure legend and methods for Fig.4 states that coronal sections were used for staining and image analysis yet the images in Fig.4 appear to be longitudinal images, please clarify this.

Reviewer #2: The reviewer appreciates the effort the authors made and thinks that the reviewers' comments have been sufficiently addressed and the data presented are sound and relevant for the field. The reviewer stull thinks that a refined analysis of the anatomical and molecular response to injury, in the form of neuronal plasticity/sprouting and cell-specific changes would add substantial value to a very interesting body of data, however acknowledges that this may be out of the scope of this particular study

7. PLOS authors have the option to publish the peer review history of their article (what does this mean?). If published, this will include your full peer review and any attached files.

Reviewer #1: No

Reviewer #2: **Yes: **Francesco De Virgiliis

---

## [Author Response · Author response to Decision Letter 1]

13 Oct 2021

RE: Resubmission of the manuscript PONE-D-21-09445R1

REVIEW RESPONSE: 

REVIEWER #1: POINT 1: The new results for the normalised horizontal ladder in both study 1 and 2 appear confusing. The mice injured at ZT12 or ZT18 do not recover significantly more but rather that the mice injured at ZT0 or ZT6 get significantly worse at 6 weeks, do the authors have any suggestions in why these groups would deteriorate at week 6?

I can appreciate why the authors tried to normalized the horizontal ladder data due to high variability. However, this is an unusual and confusing way to analyse this data and I believe sufficient number of animals were used to observe real behavioural differences between the groups. I would recommend to show the unnormalized analysis using number of errors, as was provided in the first submission.

RESPONSE: We included the “raw” error number data for both study 1 (Fig. 2B) and study 2 (Fig. 3B) as requested. The normalized data are shown as well (Fig. 2B’ and 3B’). Results description and figure legends were modified accordingly (highlighted in the track change version of the manuscript). Edits were introduced to those sections to clearly define what the normalized error score is (in the revised manuscript we referrer to that parameter as a change in error number from the previous testing). The discussion section contains a paragraph that presents the case for post-injury spasticity as a potential driver for week 6 worsening in ladder test performance of animals with daytime injuries (lines 342-353). 

REVIEWER #1, POINT 2: On Line 264 the authors state “Both BMS (A) and normalized error score in the horizontal ladder walking test (B) revealed minor yet significant improvement in locomotor recovery between ZT18 and ZT6.” But the authors go on to say that “For BMS, no significant group differences were observed with post hoc testing at any time point” Please clarify and correct the statement and results of the BMS. The data appears to show an improvement with time, due to spontaneous recovery but no differences between the groups.

RESPONSE: we apologize for those confusing statements. No significant effects of time of day on BMS were observed in study 1 or study 2. The sentences in question were all modified accordingly. 

REVIEWER #1, POINT 3: Investigating the BSCB and neuroinflammation markers is a welcomed addition to the paper. It may also be interesting to assess fluorescence intensity of these markers at the lesion site. For instance, in the image in Fig.4 CD45 staining appears to be more intense at ZT18.

RESPONSE: we agree that quantifying the BSCB/neuroinflammation markers in a region- specific manner could be potentially interesting. However, given the overall negative nature of our findings and preferable use of signal area quantifications rather than signal intensity we do not expect that such analyses would affect our conclusions in a major way. To avoid confounding effects of staining to staining variability that can affect intensity we prefer signal area analyses. Thus, the apparent difference in CD45 signal intensity is unlikely to change results of the area analyses as shown in Fig. 4B. 

REVIEWER 1, POINT 4: The figure legend and methods for Fig.4 states that coronal sections were used for staining and image analysis yet the images in Fig.4 appear to be longitudinal images, please clarify this.

RESPONSE: We use the term “coronal” to describe the sections that are cut in the coronal plane of the human body that in the spinal cord would include its long axis. We recognize that this may be confusing as the term “longitudinal” is usually used throughout the literature to describe such sections. To avoid such confusion, we replaced the term coronal with longitudinal.

---

## [Editor Report · Decision Letter 2]

21 Oct 2021

Limited changes in locomotor recovery and unaffected white matter sparing after spinal cord contusion at different times of day

PONE-D-21-09445R2

Dear Dr. Hetman,

We’re pleased to inform you that your manuscript has been judged scientifically suitable for publication and will be formally accepted for publication once it meets all outstanding technical requirements.

Kind regards,

Simone Di Giovanni

Academic Editor

PLOS ONE
---

## [Editor Report · Acceptance letter]

9 Nov 2021

PONE-D-21-09445R2 

Limited changes in locomotor recovery and unaffected white matter sparing after spinal cord contusion at different times of day. 

Dear Dr. Hetman:

I'm pleased to inform you that your manuscript has been deemed suitable for publication in PLOS ONE. Congratulations! Your manuscript is now with our production department. 

Kind regards, 

on behalf of

Dr. Simone Di Giovanni 

Academic Editor

PLOS ONE